Cancellous bone and theropod dinosaur locomotion. Part III—Inferring posture and locomotor biomechanics in extinct theropods, and its evolution on the line to birds

Bishop Peter J. 1 2 3 4 pbishop@rvc.ac.uk
http://orcid.org/0000-0003-4157-8434 Hocknull Scott A. 1 2 5
http://orcid.org/0000-0001-8174-3890 Clemente Christofer J. 6 7
http://orcid.org/0000-0002-6767-7038 Hutchinson John R. 8
http://orcid.org/0000-0002-6930-2002 Farke Andrew A. 9
Barrett Rod S. 2 3
http://orcid.org/0000-0002-0824-9682 Lloyd David G. 2 3
1 Geosciences Program, Queensland Museum , Brisbane, QLD , Australia
2 School of Allied Health Sciences, Griffith University , Gold Coast, QLD , Australia
3 Gold Coast Orthopaedic Research, Engineering and Education Alliance, Menzies Health Institute Queensland, Gold Coast , QLD , Australia
4 Current affiliation: Structure and Motion Laboratory, Department of Comparative Biomedical Sciences, Royal Veterinary College, Hatfield , Hertfordshire , UK
5 School of Biosciences, University of Melbourne , Melbourne, VIC , Australia
6 School of Science and Engineering, University of the Sunshine Coast , Maroochydore, QLD , Australia
7 School of Biological Sciences, University of Queensland , Brisbane, QLD , Australia
8 Structure and Motion Laboratory, Department of Comparative Biomedical Sciences, Royal Veterinary College , Hatfield, Hertfordshire , UK
9 Raymond M. Alf Museum of Paleontology at The Webb Schools , Claremont, CA , USA
Wedel Mathew
Electronic publication date: 2018 Oct 31
Publication date: 2018
Volume: 6
Electronic Location ID: e5777
Received 2018 Jan 8; Accepted 2018 Sep 18
Copyright: © 2018 Bishop et al.
Copyright year: 2018
Copyright holder: Bishop et al.
License: This is an open access article distributed under the terms of the Creative Commons Attribution License, which permits unrestricted use, distribution, reproduction and adaptation in any medium and for any purpose provided that it is properly attributed. For attribution, the original author(s), title, publication source (PeerJ) and either DOI or URL of the article must be cited.
License URL: https://creativecommons.org/licenses/by/4.0/

Keywords: Cancellous bone, Theropod, Bird, Locomotion, Biomechanics, Musculoskeletal modelling, Finite element modelling

Funding: An Australian Government Research Training Program Scholarship The Paleontological Society An International Society of Biomechanics Matching Dissertation Grant An Australian Research Council DECRA Fellowship DE120101503 This study was financially supported by an Australian Government Research Training Program Scholarship (to Peter Bishop), the Paleontological Society (Robert J. Stanton & James R. Dodd Award, to Peter Bishop), an International Society of Biomechanics Matching Dissertation Grant (to Peter Bishop) and an Australian Research Council DECRA Fellowship (DE120101503, to Christofer Clemente). The funders had no role in study design, data collection and analysis, decision to publish, or preparation of the manuscript.

==============================
This paper is the last of a three-part series that investigates the architecture of cancellous bone in the main hindlimb bones of theropod dinosaurs, and uses cancellous bone architectural patterns to infer locomotor biomechanics in extinct non-avian species. Cancellous bone is highly sensitive to its prevailing mechanical environment, and may therefore help further understanding of locomotor biomechanics in extinct tetrapod vertebrates such as dinosaurs. Here in Part III, the biomechanical modelling approach derived previously was applied to two species of extinct, non-avian theropods, Daspletosaurus torosus and Troodon formosus. Observed cancellous bone architectural patterns were linked with quasi-static, three-dimensional musculoskeletal and finite element models of the hindlimb of both species, and used to derive characteristic postures that best aligned continuum-level principal stresses with cancellous bone fabric. The posture identified for Daspletosaurus was largely upright, with a subvertical femoral orientation, whilst that identified for Troodon was more crouched, but not to the degree observed in extant birds. In addition to providing new insight on posture and limb articulation, this study also tested previous hypotheses of limb bone loading mechanics and muscular control strategies in non-avian theropods, and how these aspects evolved on the line to birds. The results support the hypothesis that an upright femoral posture is correlated with bending-dominant bone loading and abduction-based muscular support of the hip, whereas a crouched femoral posture is correlated with torsion-dominant bone loading and long-axis rotation-based muscular support. Moreover, the results of this study also support the inference that hindlimb posture, bone loading mechanics and muscular support strategies evolved in a gradual fashion along the line to extant birds.

Introduction

The non-avian theropod dinosaurs include some of the most recognizable of extinct animals, and with the carnivorous lifestyle and large body size of many species, they have received much attention concerning various aspects of their palaeobiology (Alexander, 1989; Bakker, 1986; Brusatte et al., 2010; Horner & Lessem, 1993; Molnar & Farlow, 1990). Locomotion in particular is a well-studied (and sometimes controversial) topic, not only because of the interest in how a giant, bipedal predator may have functioned, but also because it was likely intimately tied to the evolution of the living decendants of non-avian dinosaurs, the volant birds (Allen et al., 2013; Gatesy, 1990, 1995, 2002; Gatesy & Middleton, 1997; Hutchinson & Allen, 2009). A variety of different approaches and lines of evidence have been previouly used to address questions of locomotor biomechanics in non-avian theropods and its evolution on the line to birds, including fossil footprints (Farlow et al., 2012; Gatesy et al., 1999; Thulborn, 1990), external bone shapes and proportions (Carrano, 1998, 2000; Gatesy & Middleton, 1997; Paul, 1998), bone scaling (Carrano, 2001; Christiansen, 1999; Gatesy, 1991), midshaft cortical geometry (Alexander, 1989; Christiansen, 1998; Farlow, Smith & Robinson, 1995) and muscle attachments and significance (Carrano & Hutchinson, 2002; Gatesy, 1990; Hutchinson, 2001a, 2001b). These have been more recently augmented with various computational biomechanical models, that have examined aspects such as speed capabilities (Gatesy, Bäker & Hutchinson, 2009; Hutchinson, 2004; Hutchinson & Garcia, 2002; Sellers & Manning, 2007), muscle moment arms (Bates & Schachner, 2012; Bates, Benson & Falkingham, 2012; Hutchinson et al., 2005, 2008) and mass properties (Allen et al., 2013; Allen, Paxton & Hutchinson, 2009; Bates, Benson & Falkingham, 2012; Bates et al., 2009a, 2009b; Henderson, 1999; Henderson & Snively, 2003; Hutchinson, Ng-Thow-Hing & Anderson, 2007, Hutchinson et al., 2011).

The collective result of this prolonged and intensive research focus has been a much refined understanding of how anatomy influenced non-avian theropod stance and gait, and how these may have evolved on the line to extant birds. For instance, most non-avian species are inferred to have used a largely upright hindlimb posture during normal locomotion, where the hips and knees were flexed only to a minor degree; however, more crownward clades (e.g. paravians) may have used a more crouched posture with greater flexion at the hip and knee (Hutchinson & Allen, 2009). These postural changes are inferred to have occurred in association with changes in other biomechanically important aspects, including an anterior shift in the location of the whole-body centre of mass (COM; Allen et al., 2013), the muscular mechanisms of limb support and propulsion (Gatesy, 1990, 1995, 2002; Hutchinson & Gatesy, 2000) and bone loading mechanics (Carrano, 1998). Yet despite important advances in understanding, there is still potential for further advances to be made, from investigation of hitherto unstudied lines of evidence. One such line of evidence is the architecture of cancellous bone, which is well known from studies of extant animals to be highly sensitive and well adapted to its prevailing mechanical environment (cf. Part I of this series; Bishop et al., 2018c). Study of cancellous bone architectural patterns in non-avian theropods may therefore provide new and unique insight into various aspects of non-avian theropod locomotor biomechanics.

In Part I of this series, stark differences in hindlimb cancellous bone architecture were found between humans and birds, the only obligate bipeds alive today. Many of these differences can be associated with differences in the manner of striding, parasagittal, bipedal locomotion employed by the two groups. In particular, the differences in cancellous bone architecture reflect differences in their upright vs. crouched postures and subsequent whole-bone loading mechanics, that is, the prominence of bending and torsion. The different postures employed by humans and birds are also associated with the mechanism of muscular control required to achieve limb support during locomotion. In humans, mediolateral collapse of the stance phase limb is counteracted by hip abduction, conferred predominantly by the gluteal muscles located dorsal to the hip (Pauwels, 1980; Wall-Scheffler et al., 2010). However, in birds, anatomical, kinematic and electromyographic evidence suggests that stance limb collapse is counteracted predominantly by medial (internal) long-axis rotation of the subhorizontally oriented femur, conferred by the iliotrochantericus muscles located anterior to the hip (Gatesy, 1999b; Hutchinson & Gatesy, 2000). But what of extinct obligate bipeds, such as non-avian theropod dinosaurs?

In more stemward species of non-avian theropod, the architecture of cancellous bone in the main hindlimb bones is similar to that of humans, in terms of both principal fabric directions in the hip and knee and whole-bone architectural patterns. For instance, there exists a double-arcuate pattern in the proximal femur, roughly parallel to the coronal plane; this was not observed in more crownward non-avian species or extant birds (Part I; Bishop et al., 2018c). In species more closely related to extant birds, cancellous bone architecture tends to be more similar to that observed in birds. For instance, in the diaphysis-ward parts of the femoral metaphysis, primary fabric vectors are disorganized and often oblique to the long-axis of the bone; and in Paravians and extant birds at least, the distal tibiotarsus shows a distinctive and strongly anisotropic pattern of sagitally aligned, often plate-like trabeculae (Part I; Bishop et al., 2018c). Given that cancellous bone architectures in extant obligate bipeds appear to be linked to their different locomotor biomechanics, these observations raise the following questions regarding non-avian theropods: Did the different species of non-avian theropods employ different limb postures?

Did the bones of the different species of non-avian theropods experience different loading regimes?

Did the different species of non-avian theropods employ different strategies of muscular support in counteracting stance limb collapse?

If the different species of non-avian theropods did employ different suites of hindlimb locomotor biomechanics, how did these evolve on the line to extant birds?

Previously, the integration of anatomical, kinematic, bone strain and electromyographic data in extant species led Carrano (1998) and Hutchinson & Gatesy (2000) to hypothesize that the aforementioned aspects of bipedal locomotor biomechanics were intimately tied throughout theropod evolution. The incremental change of external osteological features throughout theropod evolution was also taken to indicate that the transformation in these particular biomechanical aspects was a gradual occurrence (Hutchinson, 2001a, 2001b; Hutchinson & Gatesy, 2000). More broadly however, the exact history of theropod locomotor evolution, in terms of whether it was long and gradual, or more punctuated at certain instances, remains to be fully discerned (Allen et al., 2013; Hutchinson & Allen, 2009).

A new approach that can quantitatively address the aforementioned questions was outlined in Part II of this series (Bishop et al., 2018b). In this ‘reverse trajectorial approach’, the observed three-dimensional (3-D) architecture of cancellous bone in the main bones of the hindlimb is coupled with musculoskeletal and finite element models of the hindlimb. Under a quasi-static situation, these models are used to derive a single ‘characteristic posture’, one in which continuum-level principal stresses best align with cancellous bone fabric. This characteristic posture is a time- and load-averaged posture across all loading regimes, and it is important to recognize that it may or may not be an actual posture used at a particular instance in a particular behaviour.

In Part II it was shown that when applied to an extant theropod (chicken, Gallus gallus), the new approach was able to retrieve a posture that was quite comparable to that used by birds at around the midstance of normal terrestrial locomotion. It could also provide a reasonable assessment of bone loading in the proximal limb (i.e. femur, proximal tibia and proximal fibula) and muscle control strategies for limb stabilization, although it had markedly lower accuracy in terms of bone loading in the distal limb (tibial shaft and below) and muscle control strategies for limb propulsion. Additionally, it was shown that the results of this approach were largely insensitive to actual muscle size (manifest as force-generating capacity), a key unknown for extinct species. When applied to extinct, non-avian theropods, the approach may therefore be used to investigate posture, bone loading mechanics and muscle recruitment patterns in these species as well. Thus, in this approach the architecture of cancellous bone constitutes an independent data set against which one or more biomechanical hypotheses may be tested.

The present study aimed to quantitatively test the hypotheses of Carrano (1998) and Hutchinson & Gatesy (2000) concerning the evolution of theropod locomotor mechanics. To do this, it applied the reverse trajectorial approach to two species of non-avian theropod, the phylogenetically basal coelurosaur Daspletosaurus torosus and the phylogenetically derived paravian Troodon formosus, to derive a single characteristic hindlimb posture that best reflects these species’ architectural patterns of cancellous bone. These species show markedly different cancellous bone architectures, with that of the former more similar to that of humans and that of the latter bearing stronger resemblance to that of birds (Part I). Understanding limb posture in these and other non-avian theropod species is in and of itself important, but it is also important for understanding other aspects of locomotion. For instance, posture can influence maximum speed capability in bipeds (Gatesy, Bäker & Hutchinson, 2009; Hutchinson, 2004; Hutchinson & Allen, 2009). In concert with the results already derived for an extant bird, the chicken (Part II), the results of this study will also facilitate an examination of how locomotor biomechanics has evolved in theropods on the line to extant birds.

Materials and Methods

The methodology employed in the present study followed that outlined previously in Part II (Bishop et al., 2018b). Essentially, musculoskeletal models of the hindlimb in a static posture were used to provide the force and boundary conditions for finite element modelling of the individual limb bones, from which principal stress trajectories were determined and compared to cancellous bone architectural patterns; the degree of correspondence between stress trajectories and cancellous bone fabric was then used to inform a new test posture. This was repeated until no further improvement in overall correspondence across the femur, tibiotarsus and fibula was able to be gained; at this point the ‘solution posture’ was achieved. Only those differences from the methodology of Part II, associated with the modelling of the two different species, will be described in the present study. Also, as with the previous study, all assumptions and model parameters were kept in their ‘best guess’ manifestation throughout the analyses; thus, differences in model results directly reflected differences in limb postures in the extinct species.

All scripts, models and data used are held in the Geosciences Collection of the Queensland Museum, and are available upon request to the Collections Manager. Additionally, a complete copy of the raw data derived from the fossil specimens is accessioned with the respective museums in which the specimens are housed.

Skeletal geometry acquisition

The models developed in this study were derived through a combination of X-ray computed tomographic (CT) scanning and photogrammetry of multiple fossil specimens; see Table 1 for the specimens (and institutional abbreviations) and imaging parameters used. The CT scans for each specimen were segmented using the software Mimics 17.0 (Materialize NV, Leuven, Belgium), via a combination of manual and automatic techniques, to produce initial surface meshes of each bone. For photogrammetry, digital photographs were taken with a Lumix DMC-TZ40 (Panasonic, Kadoma, Japan) and rendered to produce 3-D meshes using the software Agisoft Photoscan 1.0.4 (Agisoft LLC, St. Petersburg, Russia), RealityCapture 1.0 (Capturing Reality s.r.o., Bratislava, Slovakia), Meshlab 1.3.3 (http://meshlab.sourceforge.net/) and CloudCompare 2.5.4 (http://www.cloudcompare.org/).

Table 1 The specimens utilized in building the models of Daspletosaurus torosus and ‘Troodon.’

Higher-order taxonomy	Species	Specimen number*	Element	CT scan settings	
Machine	Peak tube voltage (kV)	Tube current (mA)	Exposure time (ms)	In-plane pixel resolution (mm)	Slice thickness (mm)	
Coelurosauria, Tyrannosauridae	Albertosaurus sarcophagus	TMP 81.010.0001	Pubis							
Coelurosauria, Tyrannosauridae	Albertosaurus sarcophagus	TMP 81.010.0001	Ischium							
Coelurosauria, Tyrannosauridae	Gorgosaurus libratus	TMP 1994.012.0603	Metatarsals II–IV + distal tarsals	GE Lightspeed Ultra	140	150	1,195	0.703	1.25	
Coelurosauria, Tyrannosauridae	Daspletosaurus torosus	TMP 2001.036.0001	Femur	GE Lightspeed Ultra	140	150	1,195	0.838	1.25	
Coelurosauria, Tyrannosauridae	Daspletosaurus torosus	TMP 2001.036.0001	Tibia	GE Lightspeed Ultra	120	245	1,195	0.832	1.25	
Coelurosauria, Tyrannosauridae	Daspletosaurus torosus	TMP 2001.036.0001	Fibula	GE Lightspeed Ultra	120	245	1,195	0.832	1.25	
Coelurosauria, Tyrannosauridae	Daspletosaurus torosus	TMP 2001.036.0001	Astragalus	GE Lightspeed Ultra	140	155	1,195	0.879	1.25	
Coelurosauria, Tyrannosauridae	Daspletosaurus torosus	TMP 2001.036.0001	Metatarsal IV + lateral distal tarsal	GE Lightspeed Ultra	120	185	1,195	0.738	1.25	
Coelurosauria, Tyrannosauridae	Daspletosaurus torosus	TMP 2001.036.0001	Ilium							
Coelurosauria, Tyrannosauridae	Daspletosaurus torosus	TMP 2001.036.0001	Pubis							
Coelurosauria, Tyrannosauridae	Daspletosaurus torosus	TMP 2001.036.0001	Ischium							
Coelurosauria, Tyrannosauridae	Tyrannosaurus rex	MOR 009	Metatarsal V	Toshiba Aquilion 64	135	250	750	0.625	0.5	
Coelurosauria, Tyrannosauridae	Daspletosaurus horneri	MOR 590	Metatarsals II–IV + phalanges							
Coelurosauria, Tyrannosauridae	Tyrannosaurus rex	MOR 980	Pubis							
Coelurosauria, Tyrannosauridae	Tyrannosaurus rex	MOR 980	Ischium							
Coelurosauria, Tyrannosauridae	Daspletosaurus horneri	MOR 1130	Calcaneum	Toshiba Aquilion 64	135	150	1,000	0.526	0.5	
Coelurosauria, Tyrannosauridae	Daspletosaurus horneri	MOR 1130	Metatarsal I	Toshiba Aquilion 64	135	150	1,000	0.526	0.5	
Coelurosauria, Tyrannosauridae	Teratophoneus curriei	UMNH VP 16690	Pubis							
Coelurosauria, Tyrannosauridae	Teratophoneus curriei	UMNH VP 16690	Ischium							
Paraves, Troodontidae	Latenivenatrix mcmasterae	TMP 1992.036.0575	Metatarsals II–V	Siemens Inveon	80	250	1,700	0.05	0.05	
Paraves, Troodontidae	Troodontidae sp.	MOR 553l-7.27.8.67	Ischium							
Paraves, Troodontidae	Troodontidae sp.	MOR 553s-7.11.91.41	Tibia	Siemens Inveon	80	200	1,900	0.04	0.04	
Paraves, Troodontidae	Troodontidae sp.	MOR 553s-7.28.91.239	Femur	Siemens Inveon	80	200	1,800	0.04	0.04	
Paraves, Troodontidae	Troodontidae sp.	MOR 553s-8.3.9.387	Pubis							
Paraves, Troodontidae	Troodontidae sp.	MOR 553s-8.6.92.168	Metatarsal I							
Paraves, Troodontidae	Troodontidae sp.	MOR 553s-8.17.92.265	Fibula	Siemens Inveon	80	250	1,600	0.04	0.04	
Paraves, Troodontidae	Troodontidae sp.	MOR 748	Femur	Siemens Inveon	80	200	1,900	0.04	0.04	
Paraves, Troodontidae	Troodontidae sp.	MOR 748	Tibia + astragalus + calcaneum	Siemens Inveon	80	200	1,900	0.04	0.04	
Paraves, Troodontidae	Troodontidae sp.	MOR 748	Metatarsals II–IV	Siemens Inveon	80	200	1,900	0.04	0.04	
Paraves, Troodontidae	Troodontidae sp.	MOR 748	Ilium							
Paraves, Troodontidae	Troodontidae sp.	MOR uncatalogued	Ilium							
Notes:

Also listed are the settings used in acquiring CT scans; the geometry of specimens that were not CT scanned was captured via digital photogrammetry.

* MOR, Museum of the Rockies; TMP, Royal Tyrrell Museum of Palaeontology; UMNH VP; Natural History Museum of Utah.

To maximize rigour, the models for each species were based primarily on single focal individuals that were relatively complete and well-preserved, and for which information on cancellous bone architecture was previously reported (Part I). These were TMP 2001.036.0001 for Daspletosaurus and MOR 748 for Troodon. At the time the research was undertaken, the specimens used to produce the model for Troodon were believed to represent a single species, T. formosus. However, recent research has indicated otherwise, and furthermore has cast doubt on the validity of the name T. formosus itself (Van Der Reest & Currie, 2017); the majority of specimens used in this study therefore belong to an unnamed taxon. Nonetheless, the model constructed here is still considered to be an accurate reflection of the anatomy of a large, phylogenetically derived, North American troodontid. Moreover, for the sake of simplicity in the present study, the animal being modelled will herein be referred to as ‘Troodon’.

Some bones, or parts thereof, were missing from these focal specimens, and in these cases their geometry was modelled using other specimens of the same or closely related species (Table 1). This was achieved by scaling the geometries of these other specimens appropriately to fit the focal specimens’ bones, accomplished using a combination of Mimics and the computer-aided design software Rhinoceros 4.0 (McNeel, Seattle, WA, USA). Wholesale reconstruction was required for the much of the pubis in Daspletosaurus and much of the ilium in ‘Troodon’. In Daspletosaurus, the general shape of the pubis was evident from the focal specimen, but much of the boot, pubic apron and ischiadic head were reconstructed based on comparison to other specimens that were imaged (Table 1), personal observation of other specimens in the TMP and MOR collections, and also the tyrannosaurid literature (Brochu, 2003; Osborn, 1917). In ‘Troodon’, the acetabulum, antitrochanter and pubic and ischiadic peduncles were present in the focal specimen, but the anterior and posterior iliac blades were reconstructed based on comparison to other troodontids described in the literature (Gao et al., 2012; Tsuihiji et al., 2014; Xu et al., 2002). The assembly of the individual elements of the pelvis was based on the geometry of individual bones, but also on specimens of other tyrannosaurids or paravians where the pelvic elements were preserved in situ and intact with the sacrum (Brochu, 2003; Gao et al., 2012; Lambe, 1917; Norell & Makovicky, 1997; Osborn, 1917; Tsuihiji et al., 2014; Xu et al., 2002), as well as personal observation of other specimens in the TMP and MOR collections and displays. For completeness, the vertebral column was represented by a single cylinder fixed with respect to the pelvis. In addition to the pelvis, the distalmost fibular shaft was also reconstructed for ‘Troodon’; it was essentially a continuation of the preserved part of the shaft, tapering towards the end, and gently curving laterally as it approaches the distal tibia (cf. Norell & Makovicky, 1999; Ostrom, 1969).

Some of the individual bones used in the above procedure had undergone a variable amount of taphonomic distortion. However, in all cases this appeared to be brittle deformation only, in the form of fracturing and rigid displacement of the fragments relative to one another. In these instances, the bones were retro-deformed in Rhinoceros, under the assumption of brittle deformation (Lautenschlager, 2016). This rigid retro-deformation restored the fossil geometry closer to the original geometry by realigning fragments along apposing fracture surfaces, and also taking into consideration the geometry of the bones in other specimens and other species, including comparison to the literature (Brochu, 2003; Tsuihiji et al., 2014). The retro-deformed geometries were then ‘smoothed over’ in Mimics and 3-Matic 9.0 (Materialize NV, Leuven, Belgium). Additionally, cracks or abraded edges were filled in and reconstructed in Mimics; only the minimal amount of filling in required was undertaken.

Once an initial surface mesh had been produced for the complete geometry of each bone for both species, these were smoothed in 3-matic and then refined to produce a more isoparametric mesh in ReMESH 2.1 (Attene & Falcidieno, 2006; http://remesh.sourceforge.net/). Although the tibia, astragalus and calcaneum typically remain as separate ossifications in tyrannosaurids, and the tibia remains separate from the astragalus and calcaneum in troodontids, the meshes of the three bones were fused together in this study to create a single tibiotarsus geometry. This was undertaken for the sake of simplifying the models, as well as maintaining a greater degree of consistency with the previously developed chicken model of Part II.

Musculoskeletal modelling

Musculoskeletal models of the right hindlimb of Daspletosaurus and ‘Troodon’ were constructed in NMSBuilder (Martelli et al., 2011; Valente et al., 2014) for use in OpenSim 3.0.1 (Delp et al., 2007), and are shown in Figs. 1 and 2. Both comprised 12 degrees of freedom, as in the chicken model of Part II, and 38 musculotendon actuators.

Figure 1 The musculoskeletal model of the Daspletosaurus hindlimb developed in this study.

This is shown in the ‘neutral posture’ for all joints, that is, when all joint angles are zero. (A–C) Geometries of the musculotendon actuators in relation to the bones, in lateral (A), anterior (B) and oblique anterolateral (C) views. (D–F) Location and orientation of joint coordinate systems (red, green and blue axes), the centres of mass for each segment (grey and white balls) and the soft tissue volumes used to calculate mass properties; these are shown in the same views as (A–C). Also reported in (D) are the masses for each segment; the pelvis segment represents the body as well as the contralateral limb. In (D–F), the flexion-extension axis of each joint is the blue axis. For scale, the length of each arrow in the triad of the global coordinate system is 500 mm.

Figure 2 The musculoskeletal model of the ‘Troodon’ hindlimb developed in this study.

This is shown in the neutral posture for all joints. (A–C) Geometries of the musculotendon actuators in relation to the bones, in lateral (A), anterior (B) and oblique anterolateral (C) views. (D–F) Location and orientation of joint coordinate systems (red, green and blue axes), the centres of mass for each segment (grey and white balls) and the soft tissue volumes used to calculate mass properties; these are shown in the same views as (A–C). Also reported in (D) are the masses for each segment; the pelvis segment represents the body as well as the contralateral limb. In (D–F), the flexion-extension axis of each joint is the blue axis. For scale, the length of each arrow in the triad of the global coordinate system is 200 mm.

Definition of joints

Joint locations and orientations were defined in a similar fashion to the chicken model. However, the location of the hip joint was left open-ended, so as to investigate the effects of different hip articulations (see section ‘Varying hip articulation’ below). Initially, the centre of the joint in the femur was determined by fitting a sphere to the femoral head in 3-matic, and the centre of the joint in the acetabulum was determined by positioning the centre of femoral head sphere in the centre of the acetabulum (in both lateral and anterior views). Hence, in this initial configuration, the articulation of the femur with the acetabulum was consistent with the configuration used for the chicken model. It was also consistent with the inference drawn in Part I from observations of cancellous bone architecture, that the articulation was possibly centred about the apex of the femoral head. The articulation of the tibia and fibula was guided by the relative positions of the fibular crest on the tibiotarsus and the flared anteromedial process of the proximal fibula, as well as the facet formed distally by the tibia, astragalus and calcaneum for reception of the fibula. As with the chicken model, the pes was modelled as a rectangular prism, with a width set to the mediolateral width of the distal tarsometatarsus and a length set to the total length of digit III; the total length of digit III for the ‘Troodon’ model was based on the data of Russell (1969) for Latenivenatrix mcmasterae, scaled to the individual modelled in the current study.

Definition of muscle and ligament anatomy

The hindlimb myology of Daspletosaurus and ‘Troodon’ was reconstructed through analysis of the muscle and ligament scarring patterns observed on the fossil bones, framed in the context of the myology and scarring patterns of extant archosaurs (Bates & Schachner, 2012; Bates, Benson & Falkingham, 2012; Carrano & Hutchinson, 2002; Hutchinson, 2001a, 2001b, 2002; Hutchinson et al., 2005, 2008). The 33 muscles and four ligaments reconstructed, along with their origins and insertions, are listed in Table 2. As in the chicken model, the collateral ligaments of the knee and ankle were represented by four musculotendon actuators in both the Daspletosaurus and ‘Troodon’ models. Each muscle was represented by a single musculotendon actuator in the models, with one exception; the iliotibialis 2 (IT2) was represented by two actuators on account of its probable expansive origin on the dorsal ilium (Bates, Benson & Falkingham, 2012; Hutchinson et al., 2005, 2008). The 3-D courses of the actuators were constrained to follow paths that are comparable to those reported for homologous muscles in extant archosaurs, and also as reconstructed for other non-avian theropod species (Bates & Schachner, 2012; Bates, Benson & Falkingham, 2012; Hutchinson et al., 2005, 2008).

Table 2 The origins and insertions of each of the muscles and ligaments represented in the Daspletosaurus and ‘Troodon’ musculoskeletal models.

Muscle or ligament	Abbreviation	Origin	Insertion	
Iliotibialis 1	IT1	Anterior rim of lateral ilium	Cnemial crest	
Iliotibialis 2	IT2	Dorsal rim of ilium, lateral surface	Cnemial crest	
Iliotibialis 3	IT3	Dorsal rim of postacetabular ilium	Cnemial crest	
Ambiens	AMB	Preacetabular process on proximal pubis	Cnemial crest	
Femorotibialis externus	FMTE	Lateral femoral shaft	Cnemial crest	
Femorotibialis internus	FMTI	Anteromedial femoral shaft	Cnemial crest	
Iliofibularis	ILFB	Lateral postacetabular ilium, between IFE and FTE; posterior to median vertical ridge of the ilium in Daspletosaurus	Fibular tubercle	
Iliofemoralis externus	IFE	Lateral ilium, anterodorsal to acetabulum; anterior to median vertical ridge of the ilium in Daspletosaurus	Trochanteric shelf of femur	
Iliotrochantericus caudalis	ITC	Lateral preacetabular ilium	Lesser trochanter	
Puboischiofemoralis internus 1	PIFI1	Iliac preacetabular fossa; also descending onto lateral surface of pubic peduncle in Daspletosaurus	Anteromedial aspect of proximal femur	
Puboischiofemoralis internus 2	PIFI2	Near PIFI1 origin, probably anterior to it (iliac preacetabular fossa)	Distal to lesser trochanter; on accessory trochanter in Daspletosaurus	
Flexor tibialis internus 1	FTI1	Low tubercle on posterolateral ischial shaft in Daspletosaurus; distal end of ischium in ‘Troodon’	Medial proximal tibia	
Flexor tibialis internus 3	FTI3	Ischial tuberosity on posterolateral proximal ischium in Daspletosaurus; proximal ischial shaft in ‘Troodon’	Medial proximal tibia	
Flexor tibialis externus	FTE	Lateral postacetabular ilium	Medial proximal tibia	
Adductor femoris 1	ADD1	Lateral surface of obturator process	Medial posterodistal surface of femoral shaft; large scarred region in Daspletosaurus	
Adductor femoris 2	ADD2	Posterodorsal rim of ischium	Lateral posterodistal surface of femoral shaft; large scarred region in Daspletosaurus	
Puboischiofemoralis externus 1	PIFE1	Anterior surface of pubic apron	Greater trochanter	
Puboischiofemoralis externus 2	PIFE2	Posterior surface of pubic apron	Greater trochanter	
Puboischiofemoralis externus 3	PIFE3	Lateral ischium, between ADD1 and ADD2	Greater trochanter	
Ischiotrochantericus	ISTR	Medial surface of ischium	Lateral proximal femur	
Caudofemoralis longus	CFL	Caudal vertebral centra, probably from caudal vertebrae 1–15 in Daspletosaurus and caudal vertebrae 1–10 in ‘Troodon’	Medial surface of fourth trochanter in Daspletosaurus, posteromedial surface of proximal femur in ‘Troodon’	
Caudofemoralis brevis	CFB	Brevis fossa of ilium	Lateral surface of fourth trochanter in Daspletosaurus, posterolateral surface of proximal femur in ‘Troodon’	
Gastrocnemius lateralis	GL	Posterolateral surface of distal femur	Posterior surface of metatarsals II–IV	
Gastrocnemius medialis	GM	Medial proximal tibia	Posterior surface of metatarsals II–IV	
Flexor digitorum longus	FDL	Posterior surface of distal femur	Ventral aspect of digit II–IV phalanges	
Flexor digitorum brevis	FDB	Posterior surface of metatarsals II–IV	Ventral aspect of digit II–IV phalanges	
Flexor hallucis longus	FHL	Posterior surface of femur	Ventral aspect of digit I phalanges	
Extensor digitorum longus	EDL	Distal anterolateral femur; possibly also proximal anterior tibia in Daspletosaurus.	Dorsal aspect of digit II–IV phalanges	
Extensor digitorum brevis	EDB	Anterior surface of metatarsals	Dorsal aspect of digit II–IV phalanges	
Extensor hallucis longus	EHL	Distal fibula	Dorsal aspect of digit I ungual	
Tibialis anterior	TA	Anterior surface of proximal tibia	Anterior proximal metatarsals II–IV	
Fibularis longus	FL	Anterolateral surface of tibia and/or fibula	Posterolateral ankle region (e.g. metatarsal V)	
Fibularis brevis	FB	Distal to FL on fibula	Anterolateral ankle region (e.g. metatarsal IV)	
Knee medial collateral ligament	KMCL	Depression on medial surface of medial femoral condyle	Medial proximal tibiotarsus, proximal to FCLP and FCM insertions	
Knee lateral collateral ligament	KLCL	Lateral surface of lateral femoral condyle	Lateral fibular head	
Ankle medial collateral ligament	AMCL	Depression on medial surface of astragalus	Medial proximal tarsometatarsus	
Ankle lateral collateral ligament	ALCL	Depression on lateral surface of calcaneum	Lateral proximal tarsometatarsus	
Note:

Specific differences between the two theropods are noted where appropriate.

In reconstructing the muscular and ligamentous components of the models, a number of simplifying assumptions were made. Two muscles, the ambiens (AMB) and fibularis longus (FL) may possibly have sent off secondary tendons to attach more distally in the limb, as can occur in extant archosaurs (Carrano & Hutchinson, 2002; Hutchinson, 2002). However, these secondary attachments were assumed to be of little importance for bone loading mechanics as far as the present study is concerned, and so were not modelled. A distal accessory tendon was considered to be absent from the caudofemoralis longus (CFL), as the fourth trochanter of both species lacks a distally directed process or is of small size (Carrano & Hutchinson, 2002; Hutchinson, 2001a). It is also possible that there may have been other flexor muscles of digits II–IV in both Daspletosaurus and ‘Troodon’, in addition to the flexores digitorum longus (FDL) et brevis (FDB), but currently it is too speculative to infer these (Carrano & Hutchinson, 2002; Hutchinson, 2002). It was assumed in the present study that if any such digital flexor muscles were present in either species, they would have had a similar disposition to the FDL, and so their action could be represented by the FDL actuator.

Definition of segment mass properties

To estimate the mass properties of each limb segment in the Daspletosaurus musculoskeletal model, the segment soft tissue models of Allen et al. (2013) for Tyrannosaurus were modified to fit the pelvic and limb elements of Daspletosaurus, by scaling each soft tissue segment in the x, y and z directions to fit the relevant bone or bones (and in the case of the thigh segment, to also fit the pelvis). This was accomplished in Rhinoceros. Likewise, the segment soft tissue models of Allen et al. (2013) for Velociraptor were modified appropriately to fit the pelvic and limb elements of ‘Troodon’ in the estimation of mass properties in its model. The application of the soft tissue models developed for other species to the species studied here is justified, due to close phylogenetic relationship and much similarity in the underlying skeletal structure between the species involved. Assuming a bulk density of 1,000 kg/m3 for all body segments, the total mass of the right hindlimb in the Daspletosaurus model was calculated to be 342.7 kg, and that in the ‘Troodon’ model was 5.65 kg.

To completely define the musculoskeletal model, this also required the calculation of mass properties for the remainder of the body, that is, the pelvis segment of the models. Based on femoral mid-shaft circumferences, equation 7 of Campione et al. (2014) was used to estimate the total body mass for the two models. This resulted in a mass of 2,757 kg for the Daspletosaurus model and 48.5 kg for the ‘Troodon’ model, and hence the mass of the pelvis segment in the two models (including the mass of the left hindlimb) was 2414.3 and 42.85 kg, respectively. By unintended coincidence, in both models the mass of the right hindlimb constituted approximately 12% of total body mass, which therefore increased consistency between two models. For comparison, the mass of the hindlimb in the chicken model of Part II constituted approximately 10% of total body mass. Given the data reported by Allen et al. (2013), the combined COM of the whole body, minus the right leg, in their ‘average’ model of Tyrannosaurus was 0.544 m anterior to the hip joint. The femur length of the specimen upon which their model was based is 1.265 m, as reported by Hutchinson et al. (2011). Scaling isometrically to the Daspletosaurus model, which has a femur length of 0.984 m, the COM of the pelvis segment was set at 0.423 m anterior to the hip. Similarly, the combined COM of the whole body, minus the right leg, in the ‘average’ Velociraptor model of Allen et al. (2013) was 0.090 m anterior to the hip joint, and the femur length upon which their model was based is 0.163 m. Thus, scaling isometrically to the ‘Troodon’ model, which has a femur length of 0.304 m, the COM of the pelvis segment was set at 0.168 m anterior to the hip. The dorsoventral position of the COM of the pelvis segment was assumed to be level with the hip. As noted in Part II, the dorsoventral position of the pelvis segment COM will not influence the results so long as the pelvis segment’s orientation was fixed in all simulations, and all simulations were quasi-static in nature.

Muscle activity

Not all of the 34 musculotendon actuators representing muscles were set to be active during the musculoskeletal simulations, in both Daspletosaurus and ‘Troodon’ (Table 3). The inactive muscles were set using the same criteria employed for the chicken model, and through comparison to published electromyography data for homologous hindlimb muscles in extant archosaurs (Gatesy, 1990, 1994, 1997, 1999b; Jacobson & Hollyday, 1982; Marsh et al., 2004; Reilly & Blob, 2003; Roberts, Chen & Taylor, 1998). One exception to this was the iliofemoralis externus (IFE), which in both birds and crocodilians is mostly active during the swing phase of locomotion. However, in the evolutionary scenario proposed by Hutchinson & Gatesy (2000), abductor muscles such as the IFE are expected to have been crucial to maintaining stance limb stability, if the femur was habitually held in the subvertical orientation hypothesized for most, if not all, non-avian theropods (Hutchinson & Allen, 2009). Moreover, the hypothesis of Hutchinson & Gatesy (2000) explains the stance phase inactivity of the IFE (or its homologues) in birds and crocodilians as a result of other hip muscles conferring stance limb support, namely, medial long-axis rotators in birds (iliotrochanterici) and adductors in crocodilians (adductores femoris 1 et 2). Thus, to test the hypothesis of Hutchinson & Gatesy (2000), among others, the IFE was set as being active in both the Daspletosaurus and ‘Troodon’ simulations. All active musculotendon actuators were assigned the same maximum force capacity, equal to two times body weight, that is, each muscle was capable of exerting up to 54,073.9N for Daspletosaurus and 951.2N for ‘Troodon’.

Table 3 Hypothetical activities of the muscle actuators used in the Daspletosaurus and ‘Troodon’ simulations.

Muscle	Activity	
IT1	X	
IT2	X	
IT3	X	
AMB	X	
FMTE	X	
FMTI	X	
ILFB	X	
IFE	X	
ITC	X	
PIFI1	X	
PIFI2	X	
FTI1	X	
FTI3	X	
FTE	X	
ADD1	X	
ADD2	X	
PIFE1	O	
PIFE2	O	
PIFE3	O	
ISTR	X	
CFL	X	
CFB	X	
GL	X	
GM	X	
FDL	X	
FDB	X	
FHL	X	
EDL	O	
EDB	O	
EHL	O	
TA	O	
FL	O	
FB	O	
Note:

X = active (capable of exerting up to two body weights of force), O = inactive (exerts zero force).

As in the chicken simulations of Part II, a reserve actuator was applied to the metatarsophalangeal joint in the musculoskeletal simulations. The maximum output of this actuator in the Daspletosaurus and ‘Troodon’ simulations was scaled from that set for the chicken (1,000 Nm), in proportion to the total body mass of each model: 1,767,308 Nm for Daspletosaurus and 31,090 Nm for ‘Troodon’. This corresponds to a minimum of 27 times the product of body weight and total hindlimb length (sum of interarticular lengths of femur, tibiotarsus and tarsometatarsus). By providing ample control of the metatarsophalangeal joint, this helped reduce excessively high recruitment of the FDL and FDB.

Initial posture

A general mid-stance posture was used as an initial starting point, which was modified in subsequent modelling attempts, as per the process outlined in Part II of this series. This initial posture was based on general interpretations of tyrannosaurid and troodontid appearance in the literature (technical and popular). Additionally, the hip extension angle was initially set so that the knee joint was near the line of the vertical ground reaction force in the x–z (sagittal) plane, following previous interpretations of theropod hindlimb biomechanics (Gatesy, Bäker & Hutchinson, 2009; Hutchinson & Gatesy, 2006).

Finite element modelling

Finite element simulations of the Daspletosaurus and ‘Troodon’ models were developed and solved in largely the same manner as the previously described chicken simulations of Part II, using ANSYS 17.0 (Ansys, Inc., Canonsburg, PA, USA). Two minor differences were that (i) a graduated and finer mesh was used around the cleft of the lesser trochanter of the Daspletosaurus femur, to reduce stress artifacts, and (ii) connection between the tibiotarsus and fibula entities was modelled both proximally and distally. The latter difference reflects that fact that both tyrannosaurs and troodontids possessed a distinct furrow in the distal tibiotarsus for reception of the distal fibula, whereas in birds the distal fibula is greatly reduced. In the Daspletosaurus model, the total number of elements used across the various postures tested ranged from 961,023 to 975,544 in the femur simulation and from 985,071 to 1,005,550 in the tibiotarsus + fibula simulation. In the ‘Troodon’ model, the total number of elements used across the various postures tested ranged from 668,033 to 684,547 in the femur simulation and from 583,228 to 598,556 in the tibiotarsus + fibula simulation.

Results analysis

In Part II, stress trajectories for the chicken model were compared to the observed cancellous bone architecture in birds as a whole (reported in Part I), for reasons explained there. Here, stress trajectories for the Daspletosaurus model were compared to observed cancellous bone architecture in Allosaurus and tyrannosaurid fossils, and stress trajectories for the ‘Troodon’ model were compared to observed architectural patterns in troodontid fossils. Qualitative comparisons of stress trajectories to fabric directions were made across all three bones: femur, tibiotarsus and fibula. Supplementing these qualitative assessments, quantitative comparison of stresses and architecture was undertaken for the femoral head and medial femoral condyle, followed the procedure outlined for the chicken model in Part II. The direction of minimum principal stress (σ3) was determined as the mean direction of vectors within anatomically scaled and positioned spheres placed within each region of the bone, with the mean principal fabric direction in both regions taken as previously reported in Part I (figs 22 and 29).

Varying hip articulation

Following the identification of a ‘solution posture’ for the Daspletosaurus model, a brief exploratory exercise was undertaken to address the ambiguity surrounding the articulation of non-avian theropod hips. Unlike birds, many non-avian theropods typically possessed a large incongruence in size between the femoral head and the acetabulum; for example, in the Daspletosaurus focal specimen studied, the diameter of the femoral head is about two-thirds that of the acetabulum (Fig. 3). This has consequently created uncertainty in exactly how the femur articulated with the acetabulum in these extinct species (see also Tsai & Holliday, 2015; Tsai et al., 2018). It has been previously suggested that the main area of articulation on the femur occurred on the roughly cylindrical part of the femoral head, lateral to the apex of the head (Hotton, 1980; Hutchinson & Allen, 2009). However, cancellous bone architectural patterns observed in Allosaurus and tyrannosaurids (Part I) suggest that hip joint loads may have been transmitted through the femoral head mainly from the apex of the head, not from the more lateral parts.

Figure 3 Varying the articulation of the hip joint in the Daspletosaurus model.

(A–C) The original ‘solution posture’ identified for the Daspletosaurus model. (D–F) The first variation in hip articulation, where the femur (and limb distal to it) is moved medially by 50 mm. (G–I) The second variation in hip articulation, where the femur (and limb distal to it) is moved medially by 50 mm, also with a sizeable amount of hip abduction and external long-axis rotation. (A, D and G) are in oblique anterolateral view; (B, E and H) are in close-ups of the hip articulation in anterior view; (C, F and I) show the whole hindlimb in anterior view, to illustrate the effect of differing hip articulations on gross limb position. Intervening soft tissues used in the finite element simulations are shown in turquoise; for clarity, the ilium and pubis are shown translucent in (B, E and H). Also illustrated in (B) are the relative diameters of the femoral head (solid lines) and the acetabulum (dashed lines).

To examine the effect of different hip articulations in the Daspletosaurus model, the extent of femur–acetabulum contact was varied to assess if any improvement in correspondence between principal stress trajectories and cancellous bone architecture was possible beyond that of the solution posture (Fig. 3). Two such variations were made. Firstly, the femur was moved 50 mm medially with respect to the acetabulum, so that a sizeable proportion of the cylindrical part of the femoral head was in close proximity to the acetabulum (Figs. 3D–3F). The rest of the limb was also moved medially along with the femur, including the coordinate systems of distal joints and all musculotendon actuator origins, insertions and via points that were level with or distal to the hip. So as to maintain a similar mediolateral foot placement as the original solution posture, the amount of hip abduction–adduction was altered slightly. In the second variation, the femur and limb distal to it was again moved 50 mm medially with respect to the acetabulum, but the hip was also abducted by 14°, producing a net 10° abduction from the neutral posture (Figs. 3G–3I). This reflects the amount of hip abduction that has been supposed for tyrannosaurids in previous modelling studies (Hutchinson et al., 2005; Hutchinson, Ng-Thow-Hing & Anderson, 2007), on account of the inclined disposition of the femoral head relative to the long-axis of the femur. In order to bring the foot anywhere near the body midline, this abducted posture also necessitated a large 27° of external long-axis rotation of the hip, a value comparable to maximal external long-axis rotation in modern birds during straight-line locomotion (Kambic, Roberts & Gatesy, 2015; Rubenson et al., 2007).

Cross-species patterns

Once solution postures were identified for both the Daspletosaurus and ‘Troodon’ models, a number of biomechanically relevant parameters were extracted. The same parameters were also extracted from the solution posture identified previously for the chicken model of Part II. By way of comparison across the three species, these parameters would allow a quantitative assessment of the evolutionary-biomechanical hypotheses of Carrano (1998) and Hutchinson & Gatesy (2000). Three sets of parameters were extracted: Postural parameters, related to Question 1 posed in the Introduction, the location of the whole-body COM as normalized by total hindlimb length, joint angles for the hip and knee, and the ‘degree of crouch’, both actual and predicted from empirical data reported by Bishop et al. (2018a).

Bone loading parameters, related to Question 2 posed in the Introduction, the orientation of principal stresses at the femoral mid-shaft, the ratio of maximum shear stress to bending stresses at the femoral mid-shaft, and the orientation of the neutral surface of bending at the femoral mid-shaft, relative to the mediolateral axis. To enable estimation of these parameters at mid-shaft, a local long-axis in the vicinity of the mid-point of the bone was determined. This was calculated by fitting a cylinder to the shaft in the immediate vicinity of the mid-point, using the in-built cylinder fitting tool in 3-matic; the long-axis of the cylinder defined the local long-axis of the bone, and the plane normal to this axis defined the plane of the mid-shaft cross-section. The orientation of principal stresses was defined as the orientation of the steepest inclined stress vector with respect to the local long-axis; this was calculated separately for both σ1 and σ3, and then the mean orientation was taken. In pure bending the orientation would be 0°, that is, parallel to the long-axis, and in pure torsion it would be 45° (Beer et al., 2012). Additionally, mid-shaft bending stresses were calculated as(1) σbending=|σmax|+|σmin|2

where σmax is the maximum (tensile) stress at mid-shaft and σmin is the minimum (compressive) stress at mid-shaft. This assumes that planar strain conditions were in place (Biewener, 1992), which was revealed by inspection of normal stress contours to be approximately true.

Muscular support parameters, related to Question 3 posed in the Introduction, the abduction moments of muscles that are predominantly suited to conferring hip abduction (i.e. IFE), and the long-axis rotation moments of muscles that are predominantly suited to conferring hip long-axis rotation (i.e. iliotrochantericus caudalis and puboischiofemorales internus 1 et 2 in non-avian theropods; iliotrochanterici caudalis et medialis in the chicken). To give a size-independent, dimensionless measure of how much ‘effort’ a muscle exerts to stabilize a joint about a given axis, these moments are normalized by the product of the model’s body weight and hip height:(2) M*=a⋅Fmax⋅rim⋅g⋅h

where a is the activation level of the muscle, from 0 (inactive) to 1 (maximally active), Fmax is the maximum force capable of being produced (set at two body weights as per Part II), ri is the muscle’s moment arm about joint axis i, m is body mass, g is the acceleration due to gravity (9.81 m/s2) and h is hip height. It is worth noting that this analysis carries the caveat of ignoring biarticular muscles (e.g. iliotibiales) and co-contraction between agonistic and antagonistic muscles.

Given the small sample size of species examined here (n = 3), any assessment of the evolution of biomechanically relevant parameters is necessary a coarse one. Since the hindlimb anatomy of Daspletosaurus is close to that inferred for the ancestral state of Coelurosauria, its results may taken to be reasonably representative of the most recent common ancestor of it and ‘Troodon’; likewise, since the anatomy of the ‘Troodon’ model is close to that inferred for the ancestral state of Paraves, its results may taken to be reasonably representative of the most recent common ancestor of it and the chicken. That is, it is here assumed that—in the context of locomotor biomechanics—little important evolution occurred between the ancestral coelurosaur and Daspletosaurus, and likewise little important evolution occurred between the ancestral paravian and ‘Troodon’. By mapping results towards the most recent common ancestor of successive clades, the differences observed between Daspletosaurus, ‘Troodon’ and the chicken are hence taken to be a surrogate for the actual sequence (if not pattern) of evolution along the avian stem lineage. This does not, however, escape the caveat of allometric effects on dimensional aspects of hindlimb anatomy; the issue of size effects in theropod locomotor evolution will be returned to in the ‘Discussion’.

Results

A total of five different postures for Daspletosaurus, and six postures for ‘Troodon’, were tested before no further correspondence between principal stress trajectories and cancellous bone architectural patterns was able to be achieved (Figs. 4 and 5). In the Daspletosaurus model, going from the worst to best postures tested, the angular deviation between the minimum compressive stress (σ3) and the mean direction of the primary fabric orientation (u1) in the femoral head decreased from 15.6° to 7.3°, a 53% reduction; likewise, the angular deviation between σ3 and u1 in the medial femoral condyle decreased from 11.7° to 2.8°, a 76% reduction. In the ‘Troodon’ model, going from the worst to best postures tested, the angular deviation between σ3 and u1 in the femoral head decreased from 23.8° to 3.9°, an 84% reduction; likewise, the angular deviation between σ3 and u1 in the medial femoral condyle decreased from 28.3° to 24.2°, a 14% reduction. The final solution posture for Daspletosaurus is illustrated in the centre of Fig. 4, and the solution posture for ‘Troodon’ is illustrated in the centre of Fig. 5. As with the results for the chicken model (Part II), only minimal correspondence between principal stress trajectories and cancellous bone architecture was able to be achieved in the distal tibiotarsus of either species. Little correspondence was also able to be achieved in the fibular crest of the Daspletosaurus model’s tibia. Thus, the remainder of this section will focus on the more proximal parts of the hindlimb.

Figure 4 The postures tested for in Daspletosaurus.

Around the periphery are the different postures tested, shown in lateral view, with the final solution posture in the centre box, shown in lateral, dorsal and anterior views; the whole-body COM location is also shown for the solution posture in lateral view. Joint angles for each posture are given in blue font; hip joint angles are given in the order of flexion-extension, abduction–adduction and long-axis rotation. Hip extension angle is expressed relative to the horizontal, whereas knee and ankle angles are expressed relative to the femur and tibiotarsus (respectively). For the other hip angles, positive values indicate abduction and external rotation, whereas negative values indicate adduction and internal rotation. The metatarsophalangeal joint angle is expressed relative to the neutral posture. The angular deviation between σ3 and u1 for each posture is also given in red font (reported as femoral head, then medial femoral condyle). The solution posture resulted in the greatest degree of overall correspondence between principal stress trajectories and observed cancellous bone architectural patterns in birds, as assessed by qualitative comparisons across the femur, tibiotarsus and fibula, as well as quantitative results for the femoral head and medial femoral condyle.

Figure 5 The postures tested for in ‘Troodon’.

Around the periphery are the different postures tested, shown in lateral view, with the final solution posture in the centre box, shown in lateral, dorsal and anterior views; the whole-body COM location is also shown for the solution posture in lateral view. Joint angles for each posture are given in blue font, following the same conventions as Fig. 4. The angular deviation between σ3 and u1 for each posture is also given in red font (reported as femoral head, then medial femoral condyle). The solution posture resulted in the greatest degree of overall correspondence between principal stress trajectories and observed cancellous bone architectural patterns in birds, as assessed by qualitative comparisons across the femur, tibiotarsus and fibula, as well as quantitative results for the femoral head and medial femoral condyle.

Daspletosaurus results

In the solution posture, the principal stress trajectories in the femur showed a high degree of correspondence with the observed cancellous bone architecture throughout the bone (Figs. 6 and 7). Strong correspondence between σ3 (compressive) and cancellous bone architecture occurred in the femoral head and both medial and lateral femoral condyles. This correspondence included that between the mean direction of σ3 and u1 in the femoral head (Fig. 6G) and medial femoral condyle (Fig. 7I). Correspondence between the maximum principal stress (σ1, tensile) and cancellous bone architecture occurred in the distal half of the fourth trochanter. Additionally, three instances of a double-arcuate pattern occurred, formed by σ1 and σ3, largely in the coronal plane. These correlate to three similar such patterns observed in the cancellous bone architecture of tyrannosaurids: in the femoral head and proximal metaphysis, in the lesser trochanter, and in the anterior and posterior parts of the distal femur proximal to the condyles. The double-arcuate patterns of σ1 and σ3 sometimes also occurred in the results for other postures tested, but they were often less developed compared to the solution posture.

Figure 6 Principal stress trajectories for the proximal femur in the solution posture of Daspletosaurus, compared with observed cancellous bone fabric.

For easier visual comparison, the stress trajectories were ‘downsampled’ in a custom MATLAB script, by interpolating the raw stress results at each finite element node to a regular grid. (A) Vector field of σ1 (red) and σ3 (blue) in a 3-D slice through the proximal femur, parallel to the coronal plane and through the middle of the femoral head, in anterior view. Note how the trajectory of σ3 projects towards the apex of the femoral head (green braces). (B) Observed cancellous bone architecture in the proximal femur of Allosaurus and tyrannosaurids (cf. Part I), in the same view as (A). (C) Vector field of σ1 and σ3 in a 3-D slice through the lesser trochanter, parallel to the plane of the trochanter, in anterolateral view. (D) Observed cancellous bone architecture in the lesser trochanter of Allosaurus and tyrannosaurids (cf. Part I), in the same view as (C). (E) Vector field of σ3 in the femoral head, shown as a 3-D slice parallel to the sagittal plane and through the apex of the head, in medial view. (F) Observed cancellous bone architecture in the femoral head of Allosaurus and tyrannosaurids (cf. Part I), in the same view as (E). (G) Comparison of the mean direction of σ3 in the femoral head (blue) and the estimated mean direction of u1 for Allosaurus and tyrannosaurids (red), plotted on an equal-angle stereoplot with northern hemisphere projection (using StereoNet 9.5; Allmendinger, Cardozo & Fisher, 2013; Cardozo & Allmendinger, 2013). Inset shows location of region for which the mean direction of σ3 was calculated.

Figure 7 Principal stress trajectories for the distal femur and fourth trochanter in the solution posture of Daspletosaurus, compared with observed cancellous bone fabric.

(A) Vector field of σ1 (red) and σ3 (blue) in a 3-D slice, parallel to the coronal plane and through the anterior aspect of the distal metaphysis, in anterior view. (B) Observed cancellous bone architecture in the distal metaphysis of Allosaurus and tyrannosaurids (cf. Part I), in the same view as (A). (C) Vector field of σ1 in the fourth trochanter, in medial view. (D) Observed cancellous bone architecture in the fourth trochanter of Allosaurus and tyrannosaurids (cf. Part I), in the same view as (C). (E) Vector field of σ3 in the lateral condyle, shown as a 3-D slice parallel to the sagittal plane and through the middle of the condyle. (F) Observed cancellous bone architecture in the lateral condyle of Allosaurus and tyrannosaurids (cf. Part I), in the same view as (E). (G) Vector field of σ3 in the medial condyle, shown as a 3-D slice parallel to the sagittal plane and through the middle of the condyle. (H) Observed cancellous bone architecture in the medial condyle of Allosaurus and tyrannosaurids (cf. Part I), in the same view as (G). (I) Comparison of the mean direction of σ3 in the medial condyle (blue) and the estimated mean direction of u1 for Allosaurus and tyrannosaurids (red), plotted on an equal-angle stereoplot with southern hemisphere projection. Inset shows location of region for which the mean direction of σ3 was calculated.

Strong correspondence between principal stress trajectories and cancellous bone architecture also occurred in the proximal tibia and fibula (Fig. 8). The trajectory of σ3 corresponded closely with the observed architectural patterns of both the medial and lateral condyles, including a more lateral inclination in the lateral condyle. In the cnemial crest of the tibia, the trajectory of σ1 largely paralleled the margins of the crest, as observed for cancellous bone fabric. Good correspondence between σ3 and cancellous bone architectural patterns was also observed in the fibular head, particularly for in the medial aspect of the bone (Figs. 8K and 8L).

Figure 8 Principal stress trajectories for the tibia and fibula in the solution posture for Daspletosaurus, compared with observed cancellous bone fabric.

(A) Vector field of σ3 in the medial tibial condyle, shown as a 3-D slice through the middle of the condyle and parallel to the sagittal plane, in medial view. (B) Observed cancellous bone architecture in the medial tibial condyle of Allosaurus and tyrannosaurids (cf. Part I), in the same view as (A). (C) Vector field of σ3 in the medial and lateral tibial condyles, shown as 3-D slices through the middle of the condyles and parallel to the coronal plane, in posterior view. (D) Observed cancellous bone architecture in the medial and lateral tibial condyles of Allosaurus and tyrannosaurids (cf. Part I), in the same view as (C). (E) Vector field of σ3 in the lateral tibial condyle, shown as a 3-D slice through the middle of the condyle and parallel to the sagittal plane, in lateral view. (F) Observed cancellous bone architecture in the lateral tibial condyle of Allosaurus and tyrannosaurids (cf. Part I), in the same view as (E). (G) Vector field of σ1 in the cnemial crest, shown as a 3-D slice parallel to the coronal plane, in anterior view. (H) Observed cancellous bone architecture in cnemial crest of Allosaurus and tyrannosaurids (cf. Part I), sectioned in the plane of the crest, shown in the same view as (G); blue section lines illustrate primary architectural direction. (I) Vector field of σ1 in the cnemial crest, shown as a 3-D slice parallel to the sagittal plane, in medial view. (J) Observed cancellous bone architecture in cnemial crest of Allosaurus and tyrannosaurids (cf. Part I), sectioned in the plane of the crest, shown in the same view as (I). (K) Vector field of σ3 in the medial aspect of the fibular head, in medial view. (L) Observed cancellous bone architecture in the fibular head of Allosaurus and tyrannosaurids (cf. Part I), in the same view as (K).

‘Troodon’ results

As with the Daspletosaurus model, in the solution posture identified for ‘Troodon’, the principal stress trajectories in the femur generally showed strong correspondence to the observed cancellous bone architecture (Figs. 9 and 10). Correspondence with σ3 occurred in the femoral head, under the greater trochanter and in both medial and lateral condyles; correspondence with σ1 occurred in the lesser trochanter. The mean direction of σ3 in the femoral head showed strong correspondence to the mean direction of u1 (Fig. 9E). In the medial femoral condyle, the directions of σ3 and u1 are qualitatively similar, but σ3 was notably more posteriorly inclined (by about 20°) than the mean direction of u1 (Fig. 10E), as occurred in the chicken model of Part II. Unlike the results for the Daspletosaurus model, no double-arcuate pattern of σ1 and σ3 was present in ‘Troodon’; instead, their trajectories tended to spiral about the bone’s long axis, much like the stress results for the chicken model.

Figure 9 Principal stress trajectories for the proximal femur in the solution posture of ‘Troodon’, compared with observed cancellous bone fabric.

(A and B) Vector field of σ3 in the femoral head, shown as 3-D slices parallel to the coronal plane (A, in anterior view) and sagittal plane (B, in medial view). (C and D) Observed vector field of u1 in the femoral head, in the same views as (A) and (B), respectively (cf. Part I). (E) Comparison of the mean direction of σ3 in the femoral head (blue) and the mean direction of u1 (red), plotted on an equal-angle stereoplot with northern hemisphere projection. Inset shows location of region for which the mean direction of σ3 was calculated. (F and G) Vector field of σ3 under the greater trochanter, shown as 3-D slices parallel to the coronal plane (F, in posterior view) and sagittal plane (G, in lateral view). (H and I) Observed vector field of u1 under the greater trochanter, shown in the same views as (F and G), respectively (cf. Part I). (J) Vector field of σ1 in the lesser trochanter, shown in oblique anterolateral view. (K) Observed vector field of u1 in the lesser trochanter, shown in the same view as (J) for both specimens studied (cf. Part I).

Figure 10 Principal stress trajectories for the distal femoral condyles in the solution posture of ‘Troodon’, compared with observed cancellous bone fabric.

(A) Vector field of σ3 in the lateral condyle, shown as a 3-D slice parallel to the sagittal plane. (B) Observed vector field of u1 in the lateral condyle, shown in the same view as (A) (cf. Part I). (C) Vector field of σ3 in the medial condyle, shown as a 3-D slice parallel to the sagittal plane. (D) Observed vector field of u1 in the medial condyle, shown in the same view as (C) (cf. Part I). (E) Comparison of the mean direction of σ3 in the medial condyle (blue) and the mean direction of u1 (red), plotted on an equal-angle stereoplot with southern hemisphere projection. This shows that in the solution posture the mean direction of σ3 was of the same general azimuth as the mean direction of u1, but was markedly more posteriorly inclined. Inset shows location of region for which the mean direction of σ3 was calculated.

Good correspondence between principal stress trajectories and cancellous bone architecture also occurred in the proximal tibia and fibula (Fig. 11). In the medial and lateral condyles, σ3 corresponded closely with observed architectural patterns, possessing a gentle posterior inclination, with a slight lateral inclination under the lateral condyle. In the cnemial crest, the trajectory of σ1 largely paralleled the margins of the distal part of the crest. In the fibular head, the principal stress trajectories showed good overall correspondence to the observed architectural patterns (Figs. 11K–11M). Greater correspondence occurred laterally with σ1, but some correspondence was also present in the medial side with σ3.

Figure 11 Principal stress trajectories for the tibia and fibula in the solution posture for ‘Troodon’, compared with observed cancellous bone fabric.

(A) Vector field of σ3 in the medial tibial condyle, shown as a 3-D slice through the middle of the condyle and parallel to the sagittal plane, in medial view. (B) Observed vector field of u1 in the medial tibial condyle, in the same view as (A) (cf. Part I). (C) Vector field of σ3 in the medial and lateral tibial condyles, shown as 3-D slices through the middle of the condyles and parallel to the coronal plane, in posterior view. (D) Observed vector field of u1 in the medial and lateral tibial condyles, in the same view as (C) (cf. Part I). (E) Vector field of σ3 in the lateral tibial condyle, shown as a 3-D slice through the middle of the condyle and parallel to the sagittal plane, in lateral view. (F) Observed vector field of u1 in the lateral tibial condyle, in the same view as (E) (cf. Part I). (G) Vector field of σ1 in the cnemial crest, shown as a 3-D slice parallel to the coronal plane, in anterior view. (H) Observed vector field of u1 in the cnemial crest, in the same view as (G) (cf. Part I). (I) Vector field of σ1 in the cnemial crest, shown as a 3-D slice parallel to the sagittal plane, in medial view. (J) Observed vector field of u1 in the cnemial crest, in the same view as (I) (cf. Part I). (K) Vector field of σ1 in the lateral fibular head, in lateral view. (L) Vector field of σ3 in the medial fibular head, in medial view (reversed). (M) Observed vector field of u1 in the fibular head, in the same view as (K) (cf. Part I).

Hip articulation results

In both variations in hip articulation tested for the Daspletosaurus model, the resulting principal stress trajectories of the proximal femur showed poorer correspondence with observed cancellous bone architecture than that achieved with the initial solution posture (Fig. 12). In particular, σ3, was broadly directed towards the more cylindrical part of the femoral head, lateral to the apex, rather than towards the apex itself. Additionally, the anterior inclination of σ3 in the femoral head was greater in both variations than that in the originally identified solution posture, and was markedly greater than the anterior inclination of the mean direction of u1.

Figure 12 Principal stress trajectories for the proximal femur of Daspletosaurus in the two variations in hip articulation tested.

(A) Vector field of σ3 in the first variation tested, shown as a 3-D slice parallel to the coronal plane and through the middle of the femoral head. (B) Vector field of σ3 in the first variation tested, shown as a 3-D slice parallel to the sagittal plane and through the apex of the femoral head. (C) Vector field of σ3 in the second variation tested, shown as a 3-D slice parallel to the coronal plane and through the middle of the femoral head. (D) Vector field of σ3 in the second variation tested, shown as a 3-D slice parallel to the sagittal plane and through the apex of the femoral head. (A and C) are in anterior view, (B and D) are in medial view. Note in particular how the trajectory of σ3 projects towards the more cylindrical part of the femoral head, lateral to the apex (green braces); compare to Figs. 6A, 6B, 6E and 6F. Also note in (C) how σ3 has a strong medial component near the apex of the head.

Cross-species comparisons of biomechanical parameters

In terms of posture, hip extension, hip adduction–abduction, hip long-axis rotation and knee flexion angles all changed in a gradual fashion progressing from Daspletosaurus to ‘Troodon’ to the chicken (Fig. 13). The same pattern also occurred for the anterior location of the whole-body COM and the degree of crouch. Furthermore, the degree of crouch of the solution postures matched closely with empirical predictions based on total leg length (Fig. 13C). In terms of bone loading, all parameters also changed in a gradual fashion progressing from Daspletosaurus to the chicken (Figs. 14A and 14B). Thus, in Daspletosaurus, the femur was loaded predominantly in mediolateral bending, whereas in the chicken the femur was loaded predominantly in torsion, with bending predominantly in an anteroposterior direction. In ‘Troodon’, torsion was more prominent compared to Daspletosaurus, but bending still remained the dominant loading regime. As with the other parameters, muscular support also changed gradually progressing from Daspletosaurus to the chicken (Figs. 14C and 14D). In Daspletosaurus, the normalized hip abductor moment was relatively high and the normalized hip medial rotator moment was relatively low, whereas the situation was reversed in the chicken.

Figure 13 Comparison of parameters related to posture, extracted from the solution postures of the three species modelled: Daspletosaurus (‘D’), ‘Troodon’ (‘T’) and the chicken (‘C’).

(A) Schematic illustration of the solution postures obtained for the three species, along with the location of the whole-body centre of mass (black and white disc). (B) Whole-body centre of mass location anterior to the hips, normalized to total leg length. (C) Degree of crouch for each species, both as measured from the solution posture, as well as empirically predicted from the data reported by Bishop et al. (2018a). (D) Angles of the hip and knee joints. The hip extension angle is expressed relative to the horizontal, whereas the knee flexion angle is expressed relative to the femur. (E) Long-axis rotation and adduction–abduction of the hip joint. Positive values indicate external rotation and abduction (respectively), whereas negative values indicate internal rotation and adduction (respectively).

Figure 14 Comparison of parameters related to bone loading mechanics and muscular support, extracted from the solution postures of the three species modelled: Daspletosaurus (‘D’), ‘Troodon’ (‘T’) and the chicken (‘C’).

(A) Orientation of the neutral surface of bending and the orientation of principal stresses (σ1 and σ3) relative to the femur long-axis, both measured at mid-shaft. Insets show the neutral surface with respect to the mid-shaft cross-section, as well as anatomical directions (‘A’, anterior; ‘P’, posterior; ‘M’, medial; ‘L’, lateral). (B) Ratio of maximum shear to bending stress in the femoral mid-shaft. (C) Normalized moments of hip abductor and medial rotator muscles. The hip abductor for all species is the iliofemoralis externus (activation set to zero in the chicken; see Part II). In Daspletosaurus and ‘Troodon’, the medial rotators are the iliotrochantericus caudalis and puboischiofemorales internus 1 et 2; in the chicken, they are the iliotrochanterici caudalis et medius. (D) Oblique anterolateral view of the hip of Daspletosaurus, showing the abductor and medial rotator muscles (colours as in C).

Discussion

Having previously demonstrated the validity and potential utility of the ‘reverse’ application of the trajectorial theory (Part II; Bishop et al., 2018b), the aim of the present study was to apply this approach to two extinct, non-avian theropods, Daspletosaurus torosus and ‘Troodon’ (Troodontidae sp.), to gain new insight into their hindlimb locomotor biomechanics. In addition to deriving a ‘characteristic posture’ for both species, quantitative results were produced that have bearing on various questions concerning theropod locomotor biomechanics and its evolution, posed in the Introduction. In particular, the evolutionary-biomechanical hypotheses of Carrano (1998) and Hutchinson & Gatesy (2000) were able to be quantitatively tested in a novel way.

Postures

In the ‘characteristic posture’ identified for both non-avian theropods, there was generally a strong alignment between calculated principal stress trajectories and observed patterns in cancellous bone architecture, across the femur, proximal tibia and proximal fibula. It is important to note that this should not be presumed to be the posture used by these extinct species at any particular point in the stance phase; rather, the posture identified here is a time- and load-averaged characterization of the kinds of postures experienced on a daily basis. Nevertheless, since the posture previously identified for the chicken corresponds well to the posture of a typical avian hindlimb at around mid-stance in terrestrial locomotion (Part II), the postures derived for Daspletosaurus and ‘Troodon’ are inferred to reflect the postures of these species at around the mid-stance of normal locomotion. Thus, Daspletosaurus is inferred to have stood and moved with a largely upright posture with a subvertical femoral orientation, whilst the limb posture of ‘Troodon’ is inferred to have been more crouched, although not to the degree observed in extant birds. It is worth noting that the femoral orientation of the Daspletosaurus posture, in terms of the degree of hip extension, is very similar to that hypothesized for other large, phylogenetically basal tetanuran species by previous workers such as Tyrannosaurus (Gatesy, Bäker & Hutchinson, 2009; Hutchinson, 2004; Hutchinson et al., 2005), Allosaurus and Acrocanthosaurus (Bates, Benson & Falkingham, 2012). The inferences drawn in those studies were based on the posture that allowed for high locomotor forces to be sustained (Gatesy, Bäker & Hutchinson, 2009; Hutchinson, 2004), or that achieved a maximal total moment arm of the hip extensor muscles (Bates, Benson & Falkingham, 2012; Hutchinson et al., 2005). The rationale of the latter set of studies is in some respects similar to the approach of the present study (which used static optimization in the musculoskeletal modelling stage), in that both approaches are dependent on the moment arms of individual muscles (see Part II).

Hip articulation in non-avian theropods

The results of the exploratory analysis of hip articulations in the Daspletosaurus model supported the inference made in Part I of this series: in non-avian theropods such as Allosaurus and tyrannosaurids, the immediate articulation between the femur and acetabulum may have been centred about the apex of the femoral head. Other articulations, involving greater contribution from the cylindrical part of the femoral head lateral to the apex, did not result in as strong correspondence between principal stresses and cancellous bone architecture. This is not to say that these other articulations were not used during daily activity, rather that they may have been used less frequently. Indeed, as the entire proximal surface of the non-avian theropod femur typically bears a characteristic texture indicative of a hyaline cartilage covering (smooth on the scale of millimetres, but wrinkled on the scale of centimetres; Tsai & Holliday, 2015; Tsai et al., 2018), this suggests that articulation between the lateral proximal femur and the incipient antitrochanter on the ilium would have occurred on occasion, but the relatively frequency of this remains unknown (see also Kambic, Roberts & Gatesy, 2014, 2015). This interpretation of hip articulation is also consonant with anatomical considerations of the non-avian theropod pelvis and sacrum. Specifically, a more lateral articulation of the (non-abducted) femur with the acetabulum places the femoral head more medially with respect to the pelvis, which could bring it into contact with the centra of the sacral vertebrae (Gilmore, 1920; Osborn, 1917; Rauhut & Carrano, 2016).

Combined with the results of the exploratory analysis, the solution posture identified for the Daspletosaurus model can help move towards resolving the question of how theropods with proximomedially inclined femoral heads, such as tyrannosaurids and carcharodontosaurids, kept their feet positioned close to the body midline, as indicated by fossil trackways (McCrea et al., 2014). Previously, working on the assumption that the cylindrical part of the femoral head articulated with the acetabulum, researchers had found that the femur inevitably becomes markedly abducted from the body midline. Without further speculation about joint articulations or the nature of the intervening soft tissues (cartilage, menisci) more distally in the limb, this leads to an unnaturally wide foot placement (Bates, Benson & Falkingham, 2012; Hutchinson et al., 2005, Hutchinson, Ng-Thow-Hing & Anderson, 2007). Indeed, in the second variation of hip articulation tested for the Daspletosaurus model, mediolateral step width was almost 47% of hip height (Fig. 3I), more than three times the typical step width observed in theropods (Bishop et al., 2017). With the hip articulation occurring at the apex of the femoral head, however, this allows for significant joint movement in other directions besides abduction–adduction. In particular, the solution posture identified for the Daspletosaurus model had a modest amount of external long-axis rotation, but little abduction of the femur; in fact, the femur was adducted slightly. Moreover, the asymmetry of the distal femoral condyles leads to a gently skewed orientation of the knee flexion-extension axis in the coronal plane, such that the distal crus is angled in towards the body midline (see Part II and Figs. 1E and 2E). The combination of these features allows the pes to be positioned close to the midline, yet the upper limb be kept clear of the pelvis.

Despite the potential that this new interpretation may have for understanding how non-avian theropod hips may have articulated, it is worth emphasizing that it is based on a single posture, which at best can only be regarded as a snap shot of the limb during the stance phase of locomotion. A great deal more work is required if an understanding of dynamic joint articulations throughout the stride is to be achieved. One potential avenue is by using forward dynamic simulations (Sellers et al., 2017) to generate a variety of postures throughout the stance that may be used to inform musculoskeletal and finite element models. This would require more complex modelling of some joints than is currently done (e.g. three degrees of freedom for the hip), and would in turn require substantially greater computational power.

Theropod locomotor evolution

A second major objective of the current study was to test evolutionary-biomechanical hypotheses concerning posture, bone loading mechanics and muscular control strategies in theropods. In doing so, insight would be gained as to how such aspects of theropod locomotion may have evolved on the line to birds. As only three species have thus far been investigated, current assessments are necessarily coarse; yet as these species span a broad part of the theropod family tree, this is sufficient to detect gross phyletic change in the aspects of locomotor biomechanics examined here. Indeed, that the results for the Daspletosaurus model are consistently quite different from those for the chicken model (Figs. 13 and 14) is suggestive of pronounced evolutionary change between Coelurosauria and Neognathae.

The results for the three theropod species modelled here demonstrate that, progressing through theropod phylogeny towards more derived species, the following trends occurred: The whole-body COM moved anteriorly; this was to be expected, given that model mass properties were largely derived from models developed in the study of Allen et al. (2013), who showed the same pattern.

Hindlimb posture became more crouched, at least as far as the hip and knee joints are concerned. This is consonant with the findings of previous work (Carrano, 1998; Gatesy, 1990, 1991, 1995).

Torsion became more prevalent than bending as the dominant loading regime of the femur.

The direction of bending of the femur changed from being predominantly mediolateral to being predominantly anteroposterior.

Hip abduction became overtaken by hip long-axis rotation as the main muscular control mechanism of stance-limb support.

For a given parameter, the value for ‘Troodon’ was intermediate between that for Daspletosaurus and that for the chicken. This supports the hypothesis of a gradual evolutionary change in locomotor biomechanics along the line to birds, but more taxa from different parts of theropod phylogeny would need to be modelled to definitively rule out punctuated change at any point along the stem lineage. Regardless of the mode of evolution of these parameters, the above results do suggest that hindlimb posture, bone loading mechanics and muscular support strategies were tightly associated with each other, supporting the hypotheses of Carrano (1998) and Hutchinson & Gatesy (2000). With the framework established in this series of studies, future development of models for other species, from different theropod clades, will help further test and clarify this interpretation.

The above trends identified in the present study are consilient with trends in other biomechanically relevant aspects, as noted by previous studies. These other trends include: Modifications of pelvic and hindlimb osteology and musculature (Carrano, 2000; Hutchinson, 2001a, 2001b, 2002).

Decrease in tail length and prominence of caudofemoralis musculature (Gatesy, 1990, 1995, 2002; Pittman et al., 2013).

A shift from caudofemoralis-mediated, hip-based limb retraction to ‘hamstring’-mediated, knee-based limb retraction during gait (Gatesy, 1990, 1995, 2002).

Changes in gross limb proportions, in particular a decrease in relative femur length, which in turn leads to an apparent increase in femoral diaphyseal robusticity (Carrano, 1998; Gatesy & Middleton, 1997).

The acquisition of a more continuous locomotor repertoire, where walking and running are not discrete gaits (Bishop et al., 2017).

The timing of some of these changes remains uncertain (see also Hutchinson, 2006), but it appears that all were underway prior to the origin of Paraves (i.e. birds and their closest maniraptoran relatives such as ‘Troodon’), and that many, if not all, took place over a protracted period of time.

Most of the above changes also occurred in tandem with a progressive (Lee et al., 2014) or multi-step (Benson et al., 2017) reduction in body size along the theropod stem lineage. A decrease in body size—either along the theropod stem lineage, or by directly comparing Daspletosaurus, ‘Troodon’ and the chicken—might be expected in and of itself to bring about changes in posture, since posture correlates with body size in extant parasagittal tetrapods (Biewener, 1989, 1990; Bishop et al., 2018a; Gatesy & Biewener, 1991). However, since many other aspects of theropod anatomy and locomotor biomechanics also change in tandem with body size along the theropod stem lineage, it is presently not possible to disentangle the relative importance of body size (or any other single feature) on posture. That many aspects of theropod locomotor anatomy and biomechanics appear to have co-evolved over a protracted period of time, along with additional features such as forelimb enlargement (Allen et al., 2013; Dececchi & Larsson, 2013) and elaboration of forelimb integument (Xu et al., 2014; Zelenitsky et al., 2012), is an interesting phenomenon that warrants further investigation.

The results of this study may also have more general implications for understanding locomotor biomechanics (and its evolution) in tetrapod species that employ a largely parasagittal stance and gait. Previous in vivo strain gauge studies of parasagittal mammals that use a more crouched femoral posture have shown that the femur experiences a sizeable amount of torsional loading, in addition to bending (Butcher et al., 2011; Keller & Spengler, 1989). Additionally, finite element simulations of sit-to-stand and stand-to-sit behaviour in humans, behaviours that require limb support during crouched femoral orientations, have revealed a marked increase in torsional loading of the femur compared to normal locomotion (Villette, 2016). In concert with the results of this study, these observations suggest that there is a continuum in musculoskeletal mechanics spanning from crouched to upright postures, of which birds and humans are ‘end members’. In upright postures, hip abduction is the dominant mode of limb support, which results in bending being the dominant mode of loading of the femur. However, as the femur becomes more crouched, the efficacy of hip abduction in providing limb support decreases, whilst that of hip long-axis rotation increases; this in turn loads the femur in a greater degree of torsion (see also Butcher et al., 2011).

Methodological considerations

A number of methodological considerations should be borne in mind when interpreting the results of the present study. None are considered to be of any major importance for the main interpretations made here, but they do highlight areas where future research efforts could be focused, potentially yielding further insight into theropod hindlimb biomechanics.

Correspondence in the distal tibiotarsus

It is worth re-iterating that little correspondence was evident between principal stresses and cancellous bone architecture in the distal parts of the tibiotarsus or fibula, in any posture tested for all three theropod species modelled. Additionally, the architectural patterns observed in the fibular crest of tyrannosaurid tibiae could not be replicated in the Daspletosaurus model. As discussed in Part II, this could reflect an inadequate modelling formulation, adaptation of these parts of the bones to many varied loading regimes, or a combination of both (or other) factors. For the two extinct species at least, the normal in vivo loads experienced by the distal tibiotarsus may have also been influenced by the derived arctometatarsalian structure of their metatarsus (Holtz, 1995; Snively & Russell, 2003; Wilson & Currie, 1985), a prospect requiring further investigation. Nevertheless, the architecture of cancellous bone in the distal tibiotarsus of theropods shows some strikingly different patterns between the various theropod groups. From a phenomenological perspective at least, this is indicative of marked differences in bone loading regimes, and by extension locomotor behaviour. It is therefore worthy of future modelling effort to establish a more mechanistic link between cancellous bone architecture and musculoskeletal loading mechanics in this part of the hindlimb.

Pelvic orientation

One aspect of theropod posture that was not investigated in this study was the orientation of the pelvis. In all simulations, the pelvis of the three theropod species modelled was oriented similarly, with the sacral vertebrate oriented approximately horizontally and parallel to the x-axis of the global coordinate system. However, it is known that extant birds can employ significant amounts of pitch, roll or yaw during locomotion (Abourachid et al., 2011; Gatesy, 1999a; Rubenson et al., 2007). If the pelvis underwent side-to-side rolling during locomotion in non-avian theropods, even by a small amount, this may have served to clear the pelvis and trunk further out of the way of the thigh of the stance leg. The effect of this would have been most obvious in species with well-developed pubic boots, such as large tyrannosaurids and allosauroids. Future investigation could therefore be directed towards incorporating one or more degrees of freedom in the pelvis segment of the models, as well as incorporating additional degrees of freedom in other joints (e.g. knee). Caution would need to be exercised, however, as the number of variable parameters could quickly grow to be very large, which may require a great deal more posture variations be tested before a ‘solution posture’ is satisfactorily obtained. However, as noted in Part II, the development of an automated optimization approach (in tandem with more extensive quantification of cancellous bone architecture) could allow for more degrees of freedom to be incorporated, and for more posture variations to be tested. This is a worthwhile avenue for future research, one that could make the reverse approach more easily applicable to a wider range of questions on tetrapod locomotor evolution.

Stresses in the medial femoral condyle

As noted in the results of this study, as well as those of Part II, the mean direction of the minimum principal stress (σ3) in the medial femoral condyle was notably more posteriorly inclined than the mean direction of the primary fabric orientation of cancellous bone (u1), in both the chicken and ‘Troodon’ models. This was the case regardless of the posture tested. The cause for this discrepancy is probably twofold. Firstly, taking the mean direction of u1 in the medial condyle will average out the ‘fan’ of individual fabric vectors (see Part I) that is ubiquitous in theropods. Thus, there will be some parts of the condyle for which a greater correspondence between fabric direction and the calculated principal stresses will indeed occur, namely, where the fabric vectors are more posteriorly inclined than the overall orientation.

Secondly, it is quite possible that the individual u1 vectors throughout the medial condyle may also ‘reflect’ the maximum principal stress (σ1) in addition to σ3, and so do not fully align with the calculated directions of either one. Given that motion of the theropod knee is inferred to have predominantly occurred in the flexion-extension plane (but see Kambic, Roberts & Gatesy, 2015), the main loading regimes expected in the femoral condyles would be expected be anteroposteriorly oriented, as also suggested by the ‘butterfly pattern’ of the secondary fabric direction in the condyles (see Part I). Hence, both σ1 and σ3 could be expected to be largely constrained to a parasagittal orientation, which could influence the direction of u1 throughout the medial condyle.

Conclusion

By applying the trajectorial theory in reverse, this study sought to identify a single, characteristic posture for two extinct, non-avian theropods that can explain a considerable amount of the architecture of cancellous bone observed in the hindlimb bones of these species. The postures derived for Daspletosaurus torosus and ‘Troodon’ are inferred to reflect the postures used at around mid-stance during normal terrestrial locomotion, but should not be presumed to have been the postures used. The largely upright posture identified for Daspletosaurus is comparable to the postures previously hypothesized for other large, phylogenetically basal tetanuran species of non-avian theropod. The posture identified for ‘Troodon’ is more crouched than that of Daspletosaurus, especially in regard to femoral orientation, but not to the degree observed in extant birds. The results of this study also provide an alternative perspective on the manner of articulation of the non-avian theropod hip joint, and suggest a solution to how non-avian theropods with proximomedially inclined femoral heads maintained narrow mediolateral foot placements.

In addition to improving understanding of posture in non-avian theropods, this study provides a new approach for how evolutionary-biomechanical hypotheses of locomotion can be explicitly and quantitatively tested. By using a previously underexplored line of evidence, cancellous bone architecture, the results of this study have supported the hypotheses of Carrano (1998) and Hutchinson & Gatesy (2000). Progressing from coelurosaurs through to extant birds, a number of important changes are inferred to have occurred in concert with one another, involving whole-body COM position, hindlimb posture, bone loading mechanics and muscular control strategies. The pattern of the changes also supports a more gradual fashion of change (as opposed to more punctuated), adding to the growing body of evidence suggesting that the unique locomotor repertoire of extant birds was acquired over a long period of time. However, only three species were modelled here, and so a more rigorous testing of the exact mode and tempo of evolutionary change awaits the modelling of additional species.

The integrative biomechanical modelling approach developed in Part II provides useful insights into non-avian theropod hindlimb locomotor biomechanics, as well as how this evolved along the line to extant birds. The generality of the approach means that it could be useful for understanding locomotor behaviour, and its evolution, in other extinct vertebrate groups as well. Examples of future research that could apply the approach include: forelimb posture and use in quadrupedal dinosaurs, such as ceratopsians (Fujiwara & Hutchinson, 2012; Johnson & Ostrom, 1995); the evolution of powered flight in birds, bats and pterosaurs (Bishop, 2008; Heers & Dial, 2012; Thewissen & Babcock, 1992; Unwin, 2005); the evolution of posture in synapsids on the line to mammals (Blob, 2001; Kemp, 1982; Lai, Biewener & Pierce, 2018); and the evolution of terrestrial locomotor capabilities in stem tetrapods (Clack, 2012; Pierce, Hutchinson & Clack, 2013). It may also prove to be of use for questions of biomechanics not related to locomotion, such as the posture of sauropod dinosaur necks (Stevens & Parrish, 2005; Taylor, Wedel & Naish, 2009).

The staff of the Geosciences Program of the Queensland Museum is thanked for the provision of workspace and access to literature: A. Rozefelds, K. Spring, R. Lawrence, P. Tierney, J. Wilkinson and D. Lewis. Much appreciation is extended to the staff and associated colleagues of the institutions that provided access to the material studied here: D. Henderson, B. Strilisky, G. Housego, R. Russel, T. Courtenay, B. Sanchez and F. Therrien (Royal Tyrrell Museum of Palaeontology, Drumheller); R. Irmis, C. Levitt-Bussian, C. Webb and P. Policelli (Natural History Museum of Utah, Salt Lake City); J. Horner, J. Scannella, D. Varricchio, D. Strosnider, C. Woodruff, D. Fowler and T. Carr (Museum of the Rockies, Bozeman). Many of the above people also provided helpful discussion on various aspects of theropod biology, and also helped transport specimens for CT scanning. Those who facilitated or performed the scanning itself are also greatly thanked: S. Purdy and D. Wetter (Canada Diagnostic Centres, Calgary); K. Ugrin and D. Van Why (Bozeman Deaconess Hospital, Bozeman); and S. Merchant, E. Hsu and J. Morgan (HSC Cores Research Facility, University of Utah, Salt Lake City). The thorough and constructive comments on earlier versions of the manuscript, provided by S. Gatesy, T. Ryan, D. Henderson, E. Snively and an anonymous reviewer, are all greatly appreciated, and substantially improved the clarity and content of the research presented here.

Additional Information and Declarations

Competing Interests

Author Contributions

Data Availability

John Hutchinson and Andrew Farke are Academic Editors for PeerJ.

Peter J. Bishop conceived and designed the experiments, analysed the data, contributed reagents/materials/analysis tools, prepared figures and/or tables, authored or reviewed drafts of the paper, approved the final draft.

Scott A. Hocknull conceived and designed the experiments, analysed the data, authored or reviewed drafts of the paper, approved the final draft.

Christofer J. Clemente conceived and designed the experiments, analysed the data, authored or reviewed drafts of the paper, approved the final draft.

John R. Hutchinson conceived and designed the experiments, analysed the data, authored or reviewed drafts of the paper, approved the final draft.

Andrew A. Farke analysed the data, authored or reviewed drafts of the paper, approved the final draft.

Rod S. Barrett conceived and designed the experiments, analysed the data, authored or reviewed drafts of the paper, approved the final draft.

David G. Lloyd conceived and designed the experiments, analysed the data, authored or reviewed drafts of the paper, approved the final draft.

The following information was supplied regarding data availability:

All data and code used in this series of studies collectively occupy approximately 3 Tb. It is currently impractical and prohibitively expensive to make this available via an online digital repository. As such, all data and code are held in the Geosciences Collection of the Queensland Museum, and will be made available upon a request being made to the Collections Manager (geoscience.inquiry@qm.qld.gov.au). Additionally, a complete copy of the fossil CT scan data obtained in the present study is accessioned with the respective institutions in which the specimens are housed.

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
