# Peer review of "Cancellous bone and theropod dinosaur locomotion. Part III—Inferring posture and locomotor biomechanics in extinct theropods, and its evolution on the line to birds"

_PeerJ, doi:10.7717/peerj.5777_

## Round 0.1 · original submission · Major Revisions

My decision here is virtually the same as for Part II: without discounting the comments, both pro and con, of the other reviewers, I find the concerns raised by Reviewer 1 to be worth serious consideration, particularly regarding intraspecific variation (in chickens and in Troodon) and the overlap between the non-avian theropod and bird samples. I agree that modeling a large ratite would help. It seems likely that someone is going to do that work in the near future. Is there any compelling reason why that someone should not be you?

Reviewer 1 ·

Basic reporting

The basic grammar and structure are fine. It is not clear whether the authors are going to made their raw scan data and models available.

More clarity is needed in the methods section with regards to how trabecular fabric orientation and stress orientation were compared in the two species. In Part II, the chicken stress trajectories were compared to the mean for all birds. Yet here it seems that the Troodon FEA results were compared to the Troodon fabric orientation from Part I, and the Daspletosaurus FEA results were compared to Allosaurus and tyrannosaurids? If so, this needs to be made clear in text.

Additionally, Troodon was microCT’d and subject to protocol 4 in Part I. Whereas Daspletosaurus was vet CT’d and subject to protocol 5. Which means we have quantitative data for trabecular fabric in Troodon, but qualitative data for Daspletosaurus. Given the very different sources of data, do the authors think it is then fair to incorporate these different measures of trabecular fabric into a comparative analysis and treat them as if they were equal? Furthermore, how do you make the comparisons? In Troodon, do we compare FEA stress trajectories from the “sphere of radius one-half of that fit to the entire femoral head” (Part II) for example, to trabecular orientation calculated from the same sphere, in order to find region-to-region correspondence? Could this get around the issue highlighted in Ln 830-832, that by only considering mean direction in the medial femoral condyles, you fail to account for the ‘fan’ of individual vectors?

Experimental design

Ln 123 – “appear to be linked”. The premise for this Part III would be far more convincing if a robust statistical correlation between cancellous bone architecture and locomotor biomechanics had been attempted in Part I.

Ln 146 – “a single characteristic posture”, and Ln 189 “best guess”. I question the degree to which any ‘evolutionary shifts’ in locomotor repertoire can be robustly identified, when based upon 3 models’ ‘characteristic posture’ but with no idea of intraspecific variation or modelling error.

If we refer to Figure 22B in Part I, we can see the considerable intraspecific variation present in the femoral head trabecular architecture of the chicken (in red). Indeed, the spacing between the two chicken samples appears roughly equivalent to the distance between the Troodon (Figure 22A green) and the Allosaurus + tyrannosaurids (Figure 22A purple), between which we are trying to identify these big evolutionary shifts. How confident can the authors be that these evolutionary patterns in posture would still hold given larger sample sizes and incorporating modelling error?

According to Table 1 in Part I, the authors microCT’d a second troodontid. How variable was the trabecular architecture between these two specimens? Why not repeat this modelling process with the second specimen and get a handle on how much intraspecific variation is present in the resulting ‘characteristic posture’?

Ln 328 – “modified appropriately”. Modified how? Scaled to match segment length?

Ln 423 – “solution posture”. How was this decided on? When did the authors decide that convergence had been achieved? Looking at Fig 4 A-B, it appears that angular deviation converged much more in Daspletosaurus than Troodon. Do the authors believe that further convergence genuinely could not be achieved in Troodon, even if the modelling approach was more automated to facilitate an optimisation loop between MBDA output and FEA input? It seems unjust to compare the characteristic postures of the two fossils in an evolutionary context, when the modelling protocol has converged on a solution to a greater extent in one taxa than another.

Ln 474 – How was this cylinder fitted? By eye? Least squares alignment?

Ln 515 – What postures where these? A table or a figure would be nice to visualise the suite of different postures attempted.

Ln 526 – How bad is ‘minimal correspondence’?

Ln 550 – What were the actual angular deviations in the tibia and fibula?

Validity of the findings

In this paper, much is made of testing evolutionary hypotheses regarding posture and locomotor behaviour along the theropod line. The authors' conclusions hang on interpreting potential postural and muscle activation ‘shifts’ between the 3 models, from ‘basal’ Daspletosaurus to ‘derived’ Troodon, and then the modern chicken. The authors find that Daspletosaurus had a largely upright posture, whilst Troodon “is inferred to have been more crouched, although not to be degree observed in modern birds” (ln637).

However, if we look at Figure 22 of Part I, the green dot representing Troodon’s femoral trabeculae fabric falls easily within the cluster of points representing modern birds, as does the purple area representing Allosaurus and tyrannosaurids. The same is true for Figure 29 and the medial femoral condyle. Stereoplots of the tibia are not included so I can’t comment on them. So when the authors say in Part II (ln 738) “it has been shown that birds as a whole appear to demonstrate a largely consistent pattern of cancellous bone architecture in the femur”, surely this also has to include the two fossil taxa studied in Part III?

Given that the two fossil species appear to overlap in trabecular fabric with modern birds, how is this a justification for trying to reconstruct postural evolution on the basis on trabeculae? And given that the modelling approach taken did indeed produce models of different posture, is it possible that the ‘trends’ the authors are interpreting (Ln 721-724) is actually just within the range that would characterise modern birds? I would be far more convinced of the authors findings if they also made a model of a large ratite or such in Part II. Given the findings of Bishop et al. (PLoS paper, Fig 1), the ‘postural space’ occupied by modern birds could be defined by including 2 taxa with short and long leg lengths. If the fossil taxa then fell outside that range, you could begin to interpret the fossils as being significantly different to the extant sample. This just highlights the issues with taking such a complex modelling approach, the sample size is too small to really be confident in the patterns we’re trying to interpret.

·

Basic reporting

no comment

Experimental design

no comment

Validity of the findings

nothing to add

Additional comments

This was a much easier read compared to previous two installments. Maybe my brain had absorbed enough of the large number of details of the materials and methods by this point.

I was pleased to see the general agreement between earlier anatomically inferred aspects of theropod hindlimb posture and the new quantitatively derived results. I realize that the present method is very new and that a perfect match between computed forces and bone anatomy is not likely at this early stage. The final results presented in Part III represent the outcome of a HUGE amount of work that was summarized in Parts I and II.

Some general comments about the text and figures follow, but not nearly as many as with the other two reviews.

Line 168/9: I found it a nuisance to have to dig out figure 8 from Part I. This was especially so once I saw it was just a reference to a cladogram with named nodes that I was already familiar with. It would be better to have the images immediately available.

Paragraph beginning at Line 502: large tyrannosaurids are very derived coelurosaurs and 2 orders of magnitude more massive than most maniraptorans. I think it is stretching the data too far to claim that the massive limb bones of a Daspletosaurus are representative of the primitive.ancestral state for animals like Troodon. However, the Troodon to Gallus comparisons and linking are tolerable.

Line 565: the technical term "decent" as a statement of the degree of correspondence between stresses and trabecular vectors is not sufficient. A more quantitative/statistical measure of the agreement/alignment is really needed here.

Line 660: I am puzzled how a surface can be simultaneously "smooth" and "wrinkled". Please explain.

Figure 4: I would really like to see the full, 3Dskeletal views of the intermediate postures showing the trajectory towards the solution pose. The obscured, gray stick views don't really illustrate the story.

Figures 5 - 10: I would really like to see images of the original cancellous bone immediately adjacent to the computed stress fields computed for the bone. To make room for these cancellous views I think many of the multiple views of the stress fields could be dropped.

·

Basic reporting

"Validity of the findings" and the commented manuscript have suggestions for additional references.
For Figure 3, a large anterior view of the optimal posture would better show the degree of ab/adduction of the femur. The dorsal oblique views obscure the angles. Figure 4 shows the posture for the entire hindlimb, but a close-up of the hip and femur is warranted.

Experimental design

As with the companion manuscripts, the study design is well-conceived and executed. The modeling of the distal crus and metatarsal region of Daspletosaurus is rather cartoony, but not an issue for the study's results.

Also as in the previous manuscripts, be a little careful about "basal". Traits can more or less resemble the ancestral condition, but taxa have their own specializations outside the dazzling attractor of interest (crouched limbs in extant birds). A minor issue worth just one or two slight modifications in the text.

Validity of the findings

The conclusions are certainly valid and intriguing, especially the likely sagittal orientation of Daspletosaurus's femur. I suggest adding a note and references about the arctometatarsus in "Correspondence of the distal tibiotarsus".

Additional comments

Mind-expanding results on limb posture in large theropods. The big deal here for me is how one can be a three-tonne biped, which speaks to the broad interest and applicability of the work. The series of manuscripts is a landmark of consilient methods.

---

## Round 0.2 · Minor Revisions

Congratulations, both reviewers found the manuscript much improved. There are just a handful of suggestions to deal with before the manuscript will be acceptable for publication. It will not require any further external reviews at this point.

In particular, please pay attention to Reviewer 3's suggestions for improving the clarity of the figures. They don't seem too onerous and I think they will improve the readability (and hence cite-ability) of the published paper.

Reviewer 1 ·

Basic reporting

No comment

Experimental design

No comment

Validity of the findings

No comment

Additional comments

I'd like to thank the authors for their considered and detailed response to my concerns. I do understand that the work presented comprised part of the lead author's PhD, and I am sympathetic to their limited ability to re-run analyses. I do think this research is a novel development and has a lot of promise for palaeo biomechanics.

My concerns have always stemmed from the complexity and time-consuming nature of the method, and linked to that, the relatively limited validation carried out on modern taxa. Palaeo biomechanics are often very quick to adopt exciting new methods, and sometimes I think that comes at the expense of truly understanding the processes at play in modern taxa. I have sympathy, applying the method to a T. rex is more exciting than an ostrich. However I also believe that, in terms of validation studies, if the originating authors aren't going to do it, who will? I worry that others take this method and apply it to other fossil taxa without taking the time to understand how well it works in modern species.

Ultimately, the authors have added several caveats and areas of discussion to cover these points, and the manuscript is at the point where it is acceptable for publication. It's just unfortunate that the methodological complexity and the authors' circumstances do not permit further testing on modern taxa.

·

Basic reporting

Refined presentation throughout. The anterior full-leg posture figures for the theropods are instructive. Consider more legends, labeling, and delineated clustering for some figures.

Replace neutral axis with neutral surface, the more exacting term from structural mechanics.

Clarify that M* is dimensionless, rather than an effort to divide torque by gravitational potential energy. It's an elegant way of comparing necessary musculoskeletal torques across body sizes!

Experimental design

The authors' musculoskeletal reconstructions involve a tremendous amount of well-judged effort.

Validity of the findings

The authors explain their three-taxon sample as proxies for evolutionary change, and a proof-of-concept for larger samples and more resolving or complex methods. As in the original manuscript, Daspletosaurus's small ad- and abduction of segments are revelatory for posture of large theropods.

Additional comments

The authors pay thoughtful attention to edits, suggestions, and comments on the original manuscript.

The explanation of the pelvic segment is quite clear here, in the original manuscript, and in iterations of Part II (which I missed when reviewing the Part II revision). Perhaps in relevant captions for both manuscripts, say that pelvic segment represents the mass of the body and one leg.

---

## Round 0.3 · accepted · Accept

Thank you for your diligence in addressing the concerns of the reviewers. I am satisfied with the revised manuscript, and I am happy to accept it for publication in PeerJ.

The decision of whether or not to publish the peer reviews alongside the paper is entirely yours, and will not affect how your paper is handled going forward. However, I encourage you to do so, for several reasons: (1) For this series of papers in particular, the reviewers invested considerable time and effort in providing constructive criticism. Making the reviews public allows the reviewers to receive more credit for their efforts. (2) The exchange of ideas between you and the reviewers is a valuable part of the scientific process and I'd hate to see it lost. (3) Finally, making the reviews public would contribute to the emerging culture of fairness and transparency in editing and peer review.

#